# NiH-catalysed proximal-selective hydroalkylation of unactivated alkenes and the ligand effects on regioselectivity

Xiao-Xu Wang[1,3], Yuan-Tai Xu[1,3], Zhi-Lin Zhang[1], Xi Lu [1✉] & Yao Fu [1,2✉]

Alkene hydrocarbonation reactions have been developed to supplement traditional electrophile-nucleophile cross-coupling reactions. The branch-selective hydroalkylation method applied to a broad range of unactivated alkenes remains challenging. Herein, we report a NiH-catalysed proximal-selective hydroalkylation of unactivated alkenes to access β- or γ-branched alkyl carboxylic acids and β-, γ- or δ-branched alkyl amines. A broad range of alkyl iodides and bromides with different functional groups can be installed with excellent regiocontrol and availability for site-selective late-stage functionalization of biorelevant molecules. Under modified reaction conditions with $NiCl_2(PPh_3)_2$ as the catalyst, migratory hydroalkylation takes place to provide β- (rather than γ-) branched products. The keys to success are the use of aminoquinoline and picolinamide as suitable directing groups and combined experimental and computational studies of ligand effects on the regioselectivity and detailed reaction mechanisms.

[1] Department of Hepatobiliary Surgery, The First Affiliated Hospital, Division of Life Sciences and Medicine, Hefei National Laboratory for Physical Sciences at the Microscale, CAS Key Laboratory of Urban Pollutant Conversion, Anhui Province Key Laboratory of Biomass Clean Energy, University of Science and Technology of China, 230026 Hefei, China. [2] Institute of Energy, Hefei Comprehensive National Science Center, 230031 Hefei, China. [3] These authors contributed equally: Xiao-Xu Wang, Yuan-Tai Xu. ✉email: luxi@mail.ustc.edu.cn; fuyao@ustc.edu.cn

The construction of molecular skeletons by C($sp^3$)-centred coupling under mild conditions has been a goal in organic synthesis chemistry for a long time[1,2]. Increasing the number of saturated carbon centres is of positive significance to the success of clinical drug research[3]. Traditional C($sp^3$)-centred cross-coupling relies on alkyl metallic reagents[4–6]. However, commercial approaches to access alkyl metallic reagents are limited. The preparation of these reagents in the laboratory is time-consuming, and the operation under anhydrous and anaerobic harsh conditions is inconvenient[7]. In recent years, alkene hydrocarbonation reactions have been developed to supplement traditional electrophile-nucleophile cross-coupling reactions[8,9]. The in situ generation of a catalytic organometallic intermediate from alkenes instead of a stoichiometric metallic reagent to participate in C($sp^3$)-centred coupling simplifies the synthesis steps and routes and improves the functional group compatibility. Alkene hydroarylation(alkenylation) with simple arenes (ArH), aryl electrophile (ArX), or aryl nucleophile (ArM) delivered corresponding alkene addition products through various mechanistic pathways[10–18]. Remarkable advances were achieved in both linear- and branched-selective hydroarylations[19–28]. On the other hand, alkene hydroalkylation, enabling the rapid construction of alkyl-alkyl bonds, would broaden our horizon of retrosynthetic analysis when facing the pervasive C($sp^3$)–C($sp^3$) bonds in organic molecules[1,2]. Hydroalkylation of terminal alkenes has enjoyed success[10,21,29–36]. Increasing attention has been given to the hydroalkylation of internal alkenes, especially nickel-hydride catalysed processes (Fig. 1a). However, nickel-hydride migratory insertion is sensitive to alkene substrates' steric and electronic properties. The rate of hydrometalation of unactivated internal alkenes is much slower than that of terminal alkenes[37–39]. Chain-walking isomerization often occurs in internal alkenes and causes linear-functionalized products[40,41]. Another successful approach is the hydrofunctionalization of activated (conjugated) alkenes[42]. The Lewis basic oxygen in the directing group or conjugated aryl or boryl group effectively stabilizes the alkylnickel intermediate, obtaining α-alkylated products with high regioselectivity[43–46]. A report on NiH-catalysed hydroalkylation of α,β-unsaturated amides to access β-alkylated products was also recently disclosed[47]. Despite the success of linear-selective migratory hydroalkylation of internal alkenes and

α- or β-selective hydroalkylation of activated (conjugated) alkenes, the branch-selective hydroalkylation method applied to a broad range of unactivated alkenes remains challenging.

Amide-linked directing groups, such as aminoquinoline (AQ) and picolinamide (PA), are widely used in palladium-catalysed C–H functionalization[48–52]. In recent years, the directing group concept has been transplanted to the research field of alkene functionalizations[53–56]. Many functional groups, including boron, amine, aryl, and alkyl groups, have been successfully installed into the double bonds of unactivated alkenes using palladium or nickel catalysts[21,22,57–62]. For example, Engle and co-workers[63] and Koh and co-workers[64,65] independently reported the dicarbofunctionalization of nonconjugated alkenes with aminoquinoline directing groups. The directing group concept also opens up new directions for the NiH-catalysed hydrofunctionalization of unactivated internal alkenes[15,23,66,67]. Recently, Koh and co-workers made progress in NiH-catalysed branch-selective hydroalkylation of unactivated alkenes using the aminoquinoline directing auxiliary. Alkene migratory hydroalkylation undergoes smoothly to access β-alkylated amides with excellent regioselectivity[68]. From the perspective of the previous literature, hydroalkylation of internal alkenes on a remote position to functional units (rather than α- or β-alkylation) remains challenging.

Herein, we report a NiH-catalysed proximal-selective hydroalkylation of unactivated alkenes to access β- or γ-branched alkyl carboxylic acids and β-, γ- or δ-branched alkyl amines. Using NiCl$_2$(PPh$_3$)$_2$ as the catalyst, migratory hydroalkylation takes place to provide β- (rather than γ-) branched products (Fig. 1b). A preliminary result with moderate enantioselectivity demonstrates that an asymmetric variant can be realized. The keys to success are the application of aminoquinoline and picolinamide as suitable directing groups and detailed studies of the ligand effects and reaction mechanisms.

## Results and discussion

**Reaction discovery**. We commenced this study with alkene **1a** and alkyl iodide **2a** as the model substrates (Table 1). Amide-directed proximal-selective hydroalkylation was efficiently

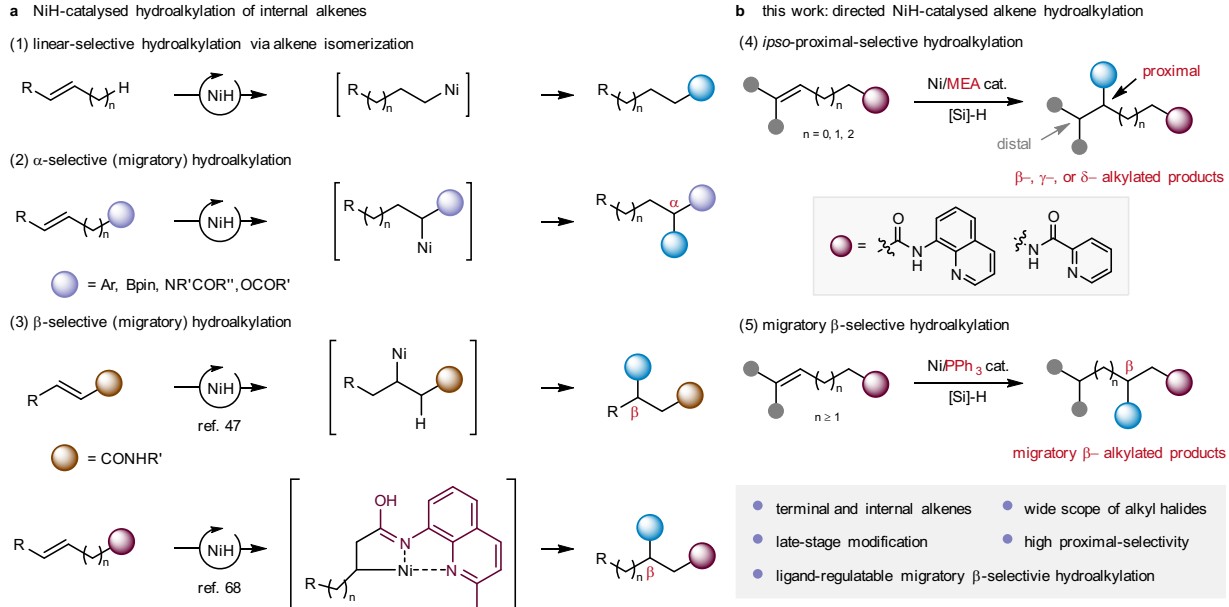

**Fig. 1 NiH-catalysed hydroalkylation of internal alkenes and their regioselectivities. a** NiH-catalysed linear-, α- or β-selective (migratory) hydroalkylation of internal alkenes. **b** Our strategy: directed NiH-catalysed proximal-selective hydroalkylation and migratory β-selective hydroalkylation. Bpin pinacol borate, diglyme 2-methoxyethyl ether, MEA ethanolamine.

**Table 1 Optimization of the reaction conditions.**

| Entry | Variation from standard conditions A | Yield/%[a] | r.r.[b] |
|---|---|---|---|
| 1 | None | 90 (87[c]) | >20:1 |
| 2 | NiBr$_2$(DME) instead of NiBr$_2$(diglyme) | 85 | 18:1 |
| 3 | Ni(acac)$_2$ instead of NiBr$_2$(diglyme) | 79 | 14:1 |
| 4 | NiCl$_2$(PPh$_3$)$_2$ instead of NiBr$_2$(diglyme) | 48 | 1.1:1[d] |
| 5 | (MeO)$_3$SiH instead of DEMS | 80 | 5.4:1 |
| 6 | PMHS instead of DEMS | 77 | 18:1 |
| 7 | Ph$_2$SiH$_2$ instead of DEMS | 72 | >20:1 |
| 8 | MeEt$_2$SiH instead of DEMS | N.R. | — |
| 9 | Na$_2$CO$_3$ instead of KF | 80 | 3.5:1 |
| 10 | CsF instead of KF | 34 | >20:1 |
| 11 | NaHCO$_3$ instead of KF | 57 | 6.2:1 |
| 12 | K$_3$PO$_4$(H$_2$O) instead of KF | 87 | 7.0:1 |
| 13 | DMF instead of DMAc | 81 | 17:1 |
| 14 | MeCN instead of DMAc | 25 | 12:1 |
| 15 | DCE instead of DMAc | 33 | 10:1 |
| 16 | Toluene instead of DMAc | 20 | 8.1:1 |
| 17 | THF instead of DMAc | 12 | 1.0:1 |
| 18 | w/o NiBr$_2$(diglyme) | N.R. | — |

Reactions were carried out under an argon atmosphere. Conditions: **1a** (0.10 mmol, 1.0 equiv), **2a** (0.20 mmol, 2.0 equiv), nickel catalyst (0.01 mmol, 10 mol%), MEA (0.012 mmol, 12 mol%), silane (0.30 mmol, 3.0 equiv), base (0.30 mmol, 3.0 equiv), solvent (0.50 mL, 0.2 M), 12 h. Proximal-selective hydroalkylation product **4aa** was obtained as the major regioisomer, and distal-selective hydroalkylation product **4aa'** as the minor regioisomer; other regioisomers could hardly be detected.
*DMF* N,N-dimethylformamide, *DME* 1,2-dimethoxyethane, *THF* tetrahydrofuran, *acac* acetylacetanoate, *w/o* without, *N.R.* no reaction.
[a]Yields and regioisomeric ratios were determined by GC analysis with triphenylmethane as an internal standard. Total yield for the mixture of all regioisomers.
[b]r.r. refers to the regioisomeric ratio, that of the major product to the sum of all other isomers.
[c]Isolated yield in parentheses.
[d]A large amount of β-selective product was observed by GC analysis.

performed using NiBr$_2$(diglyme) as the catalyst, MEA as the ligand, DEMS as the hydride source, KF as the base in a DMAc solvent (entry 1), and desired product **4aa** was obtained in 90% GC (gas chromatography) yield, 87% isolated yield, and >20:1 r.r. (regioisomeric ratio). NiBr$_2$(DME) and Ni(acac)$_2$ could be used instead of NiBr$_2$(diglyme) as catalysts (entries 2–3), but NiCl$_2$(PPh$_3$)$_2$, which would introduce a phosphine ligand into the reaction system, led to a dramatic negative effect on regioselectivity (entry 4). Oxygen-bearing silanes, such as (MeO)$_3$SiH and PMHS (entries 5–6), gave good yields; Ph$_2$SiH$_2$ also performed well (entry 7), but the less electrophilic hydrosilane MeEt$_2$SiH was inefficient in this transformation (entry 8). KF turned out to be the best base over the other bases (entries 9–12), such as Na$_2$CO$_3$, CsF, NaHCO$_3$, and K$_3$PO$_4$(H$_2$O), because of its matched ability to activate DEMS to form pentacoordinate hydrosilicate species. We could regulate the reaction activity through the cross-combination of different bases and silanes. This reaction is sensitive to solvent identity (entries 13–17); an excellent yield was obtained in DMF, which is an amide solvent (entry 13), while poor results were obtained using MeCN, DCE, or toluene (entries 14–16), and disappointing regioselectivity was obtained when using THF as the solvent (entry 17). Finally, this reaction was completely shut down in the control experiment without a nickel catalyst (entry 18).

**Substrate scope.** With appropriate reaction conditions in hand, we sought to examine the substrate scope of this amide-directed nickel-catalysed hydroalkylation. As summarized in Fig. 2, a wide range of alkyl halides and alkenes was examined and provided satisfying coupling yields (40–88%) and regioselectivities (4.0:1~>20:1) in almost all cases. Both alkyl iodides **2a-2l**, **2n**, **2q-2s** and bromides **3a**, **3m**, **3o**, **3p**, and **3t** were suitable substrates, and 0.5 equiv NaI was added for the in situ generation of the corresponding alkyl iodides from alkyl bromides. The β-branched alkyl halides (**2b** and **2c**) delivered products in slightly diminished yields and equally good regioselectivities. Concerning alkenes, both terminal (**1a**) and internal alkenes with differentiated steric resistances (**1b–1g**, **1i–1l**) could be used. The Z/E configuration of the internal alkene would not affect the coupling yield or regioselectivity (**4ba**). In addition, the trisubstituted alkene (**1g**), which is difficult for nickel-hydride insertion, could be transformed in a moderate yield. An interesting example was **4la**. The alkene substrate **1l** with a potentially competitive activated benzylic reaction site delivered the desired amide-directed γ-alkylation product with high regioselectivity, indicating that the aminoquinoline auxiliary displayed stronger directing ability compared to an aromatic ring. In another example, the derivative of but-3-enoic acid (**1m**) was selectively converted to the β-branched product **4ma** following proximal hydroalkylation selectivity. Finally, picolinamide as an auxiliary also produced gratifying proximal hydroalkylation results (β-alkylation product **4na**, γ-alkylation product **4oa**, and δ-alkylation product **4pa**) under standard conditions. However, the moderate regioisomeric ratios (4.0:1~4.8:1) need further reaction condition optimization. A variety of functional groups were well accommodated, such as trifluoromethyl (**4ad**), ester (**4ae**), ether (**4af**, **4ag** and **4ea**), and cyano (**4ah**, and **4ai**) groups. Under mild reaction conditions,

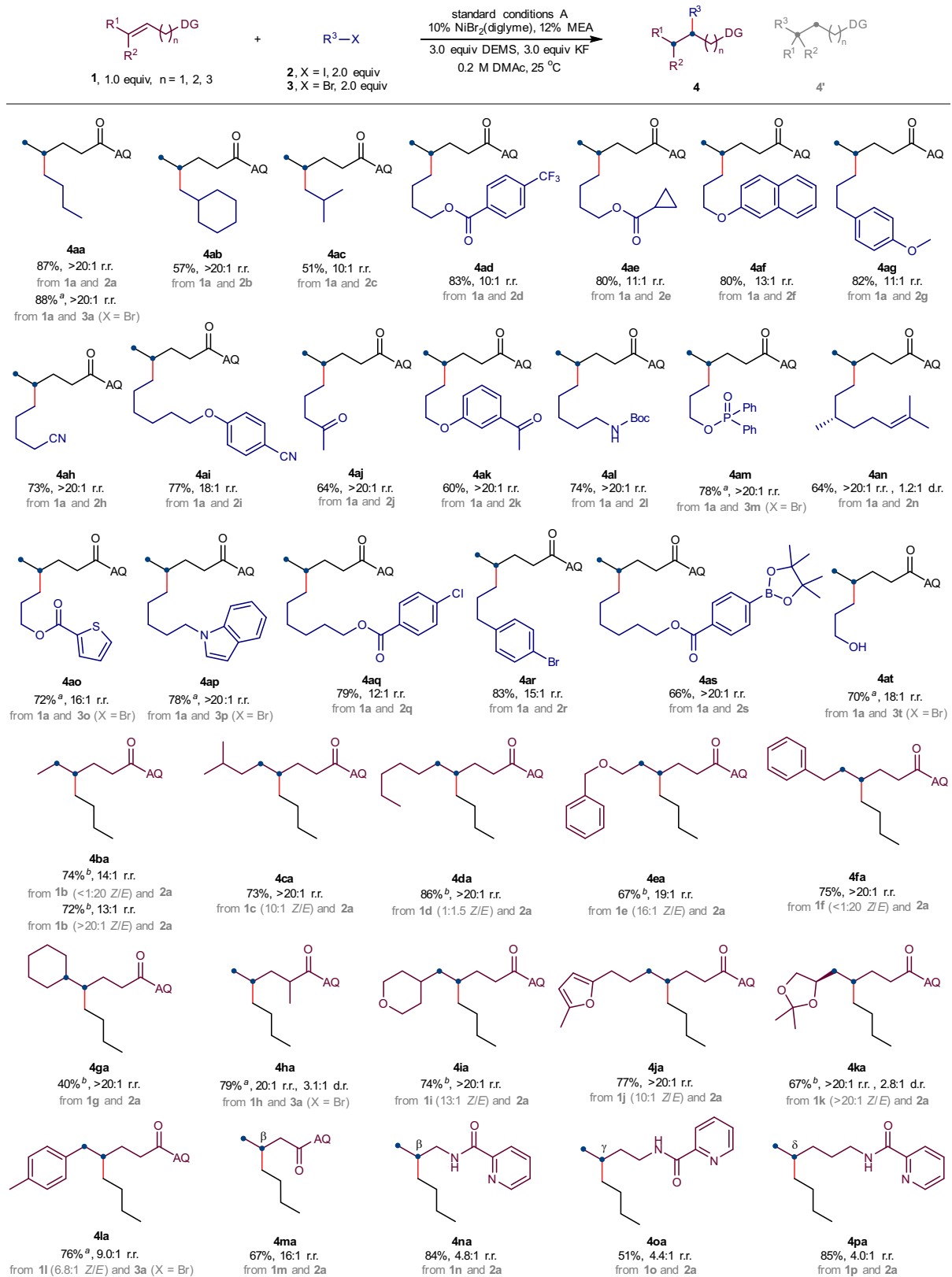

**Fig. 2 Scope of proximal-selective hydroalkylation.** Standard conditions A: alkene (0.20 mmol, 1.0 equiv), alkyl halide (0.40 mmol, 2.0 equiv), NiBr$_2$(diglyme) (0.02 mmol, 10 mol%), MEA (0.024 mmol, 12 mol%), DEMS (0.60 mmol, 3.0 equiv), KF (0.60 mmol, 3.0 equiv), DMAc (1.0 mL, 0.2 M), 25 °C, 12 h, isolated yield. r.r. was determined by GC or $^1$H NMR analysis. r.r. refers to the regioisomeric ratio, that of the major product to the sum of all other isomers. Proximal-selective hydroalkylation product was obtained as the major regioisomer, and distal-selective hydroalkylation product as the minor regioisomer; other regioisomers could hardly be detected. $^a$NaI (0.10 mmol, 0.5 equiv) was added. $^b$40 °C. d.r. diastereomeric ratio, DG directing group.

many sensitive functional groups could be tolerated, including base-sensitive ketones (**4aj** and **4ak**) and acid-sensitive *tert*-butoxycarbonyl (**4al**) and propylidene (**4ka**) protecting groups. Phosphonate (**4am**) and internal alkenyl groups (**4an**) were compatible during the transformation. Heterocyclic compounds such as thiophene (**4ao**), indole (**4ap**), tetrahydropyran (**4ia**), and furan (**4ja**) posed no problems. Electrophilic aryl chloride (**4aq**) and aryl bromide (**4ar**), as well as nucleophilic arylboronate (**4as**) survived, providing the possibility for further transformation through transition-metal-catalysed cross-coupling reactions. Finally, this reaction could be conducted in the presence of an unprotected hydroxyl group (**4at**), demonstrating its high degree of compatibility with sensitive functional groups. We examined several secondary and tertiary electrophiles (see Supplementary Table 5 and Supplementary Fig. 20 for more details). For the reaction using iodocyclohexane as an electrophile, the hydroalkylation product was delivered in a 23% yield (2.4:1 r.r.) along with the alkene protonation byproduct in a 38% yield; slight modification using (MeO)$_3$SiH as hydride source gave out a 35% total yield (1.9:1 r.r.). For the reaction using tertiary electrophiles, such as 2-iodo-2-methylpropane, 2-bromo-2-methylpropane, and 1-bromoadamantane, the desired hydroalkylation products were not observed. In the example of trisubstituted alkene **4ga**, a secondary-primary C($sp^3$)-C($sp^3$) bond formed with a moderate yield. However, the attempted construction of a tertiary-primary C($sp^3$)-C($sp^3$) bond failed in the examination of 1,1-disubstituted alkene and tetrasubstituted alkene (see Supplementary Fig. 21 for more details). Therefore, this reaction might be sensitive to the steric hindrance of both alkyl halides and alkene substrates.

**Synthetic applications**. We explored the usefulness of this hydroalkylation reaction as an efficient method for the late-stage modification of biorelevant molecules (Fig. 3a). Alkyl halides derived from drug molecules and natural products, such as indomethacin, lithocholic acid, methylumbelliferone, and D-galactopyranoside, underwent the hydroalkylation process smoothly with high chemoselectivity and good yields. The tolerance of aryl chloride, acetal groups, and unprotected alcohol hydroxyl groups highlights the excellent compatibility with sensitive functional groups. In a scale-up reaction (Fig. 3b), we achieved alkene hydroalkylation product **4aa** with a satisfactory 78% isolated yield and 12:1 regioselectivity. Under mild nickel-catalysed conditions, the directing amide group was transformed into a methyl ester group to deliver the desired methyl carboxylate **5a** in a 70% isolated yield. After hydrolysis of **4aa** in hydrochloric acid solution, alkyl carboxylic acid **5b** was obtained in a 73% isolated yield with 96% aminoquinoline recovered. The gram-scale reaction and successful removal of the directing group highlight the practicality of this nickel-catalysed alkyl-alkyl coupling reaction.

**Mechanistic studies**. We carried out a series of control and mechanistic experiments to elucidate the reaction mechanism (Fig. 4). In the radical clock experiment (Fig. 4a) using (bromo-methyl)cyclopropane (**3y**) as an electrophile, coupling product **4ay** originating from the ring-opening of the cyclopropylmethyl radical was detected. When 6-iodohex-1-ene (**2z**) was subjected to the standard conditions, we observed that the expected ring-cyclized product **4az** originated from the cyclization of the 5-hexenyl radical. These data support the radical nature of this reductive hydroalkylation reaction. In the isotope labelling experiment (Fig. 4b) with Ph$_2$SiD$_2$ (97% deuterium content) under standard conditions, deuterium was incorporated exclusively at the terminal position with a 95% deuterium content, indicating that hydrogen is provided by hydrosilane via the generation of NiH species. In addition, we experimented with δ,δ-dideuterated alkene $d_2$-**1a**. The δ-position

**Fig. 3 Synthetic applications. a** Late-stage functionalization of biorelevant molecules. **b** Gram-scale reaction and removal of the directing group. Standard conditions A: as shown in Fig. 2, 0.2 mmol scales. All yields refer to the isolated yield of purified products. Regioisomeric ratios and diastereomeric ratios were determined by GC or $^1$H NMR analysis. tmhd 2,2,6,6-tetramethyl-3,5-heptanedionato.

deuterium atoms in the dideuterated substrate remained intact, and no H/D exchange was observed along the alkyl chain. Next, we examined the influence of nitrogen-containing ligands on the coupling yield and regioselectivity (Fig. 4c). This reaction could be conducted in the absence of ethanolamine ligand or by using ethy-lenediamine (**L1**), butylamine (**L2**), or aniline (**L3**) as the ligand. However, diminished 5.3:1~7.8:1 regioisomeric ratios demonstrated the significance of ethanolamine (MEA) in regulating the regios-electivity. Moreover, bidentate bipyridine (**L4**) and tridentate ter-pyridine (**L5**) ligands showed a conspicuous decline in coupling efficiency and regioselectivity. This could be attributed to the exces-sive occupation of the coordination space on the nickel centre. In a preliminary experiment using chiral diamine ligand **L6** (Fig. 4d), this proximal-selective hydroalkylation reaction could be realized with moderate enantioselectivity (see Supplementary Tables 2 and 3 for more details). More efforts are needed to improve enantioselectivity. Note that chiral diamine ligands, including but not limited to **L6**, could be conveniently synthesized from the corresponding imines

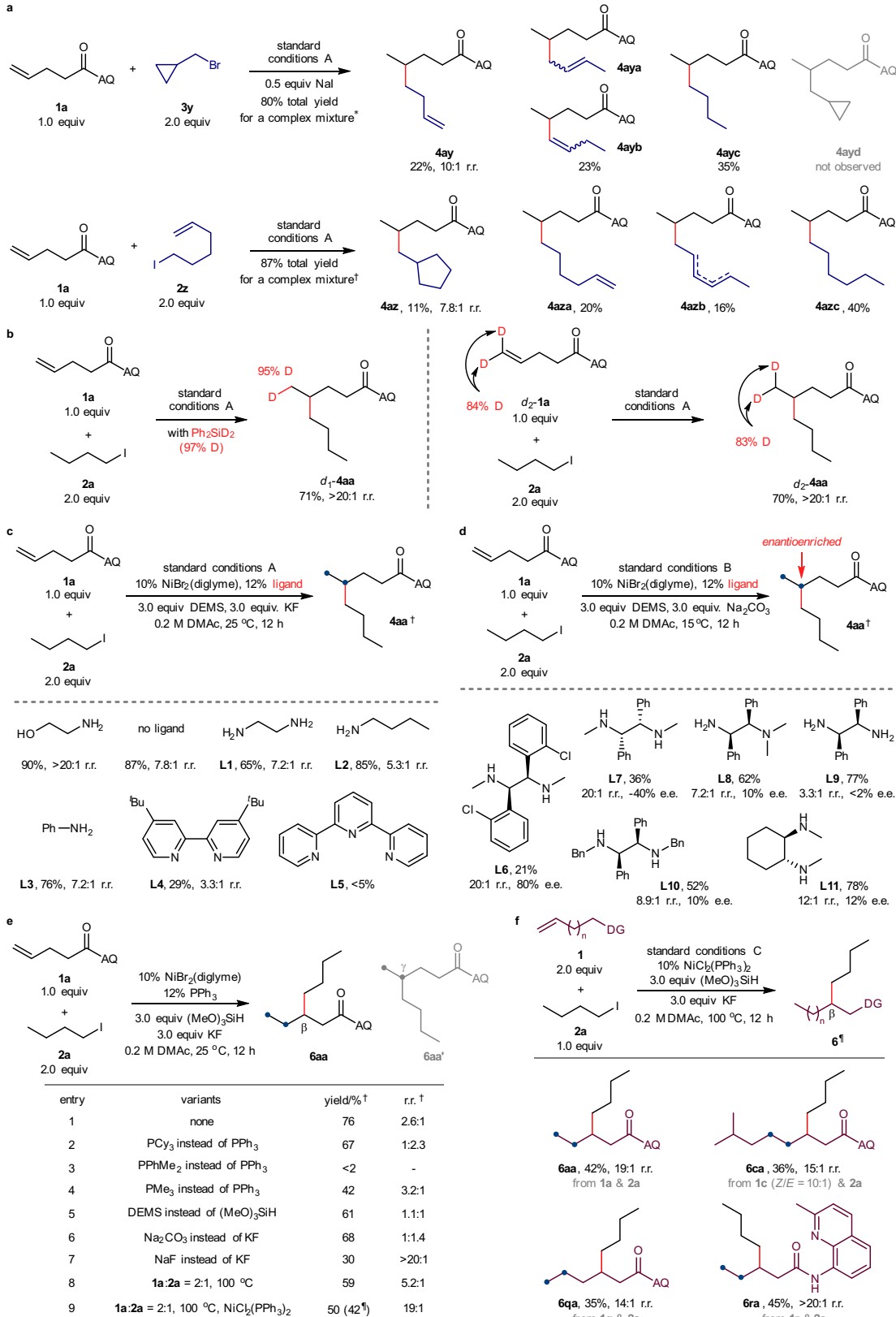

**Fig. 4 Mechanistic studies. a** Radical clock experiments. **b** Deuterium labelling experiments. **c** Effects of nitrogen-containing ligands. **d** Preliminary results of asymmetric synthesis. **e** Effects of phosphine-containing ligands. **f** Migratory β-selective hydroalkylation. Standard conditions A: as shown in Fig. 2, 0.2 mmol scales. Standard conditions B: determination of regioisomeric ratio was consistent with standard conditions A, 0.1 mmol scales. Standard conditions C: total yield for the mixture of all regioisomers. r.r. refers to the regioisomeric ratio, that of the major product to the sum of all other isomers. In most cases, a mixture of β- and γ-selective products was obtained; we could hardly observe other isomers. *[1]H NMR yield with dibromomethane as an internal standard. †GC yield with triphenylmethane as an internal standard. ¶Isolated yield in parentheses.

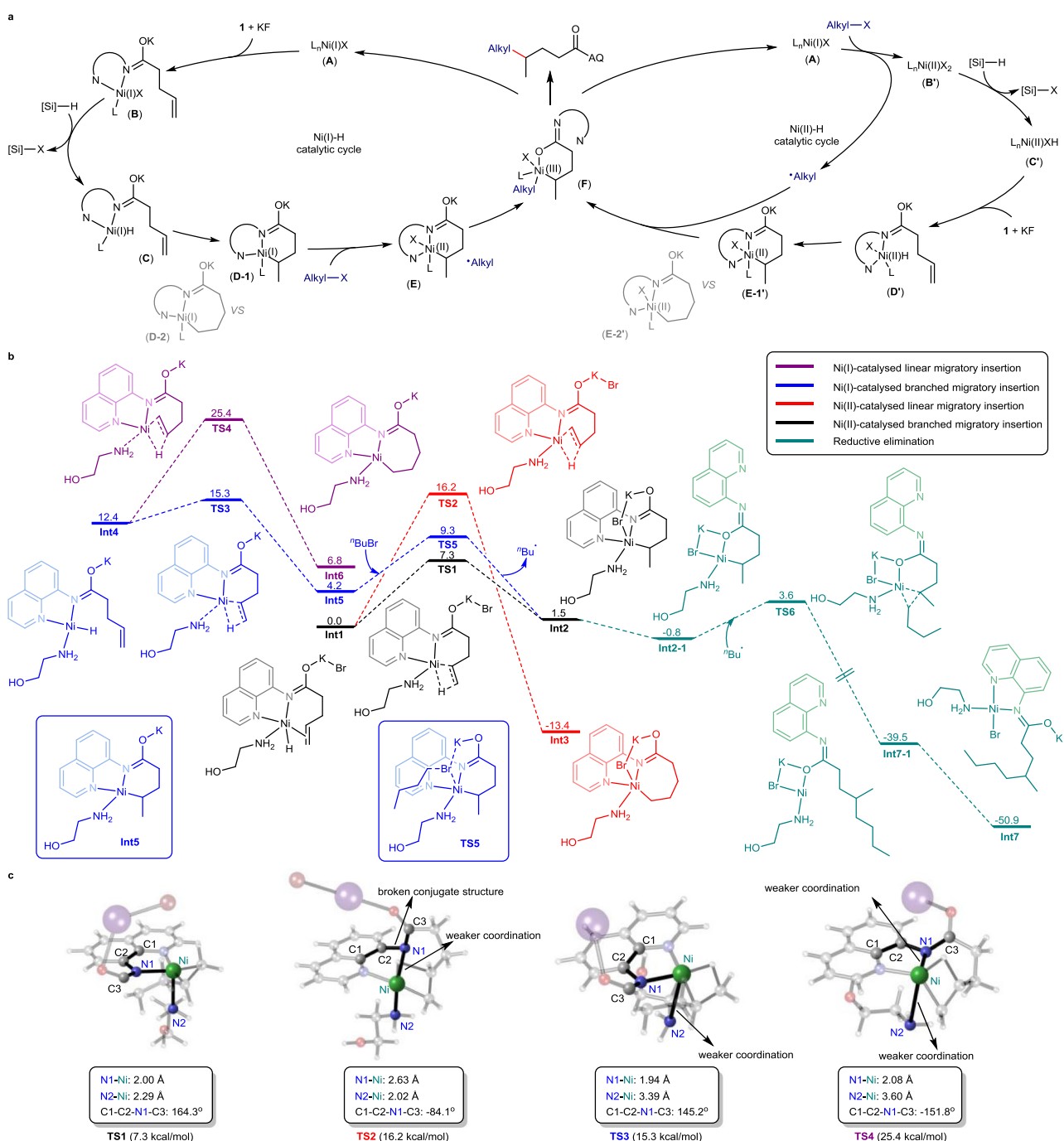

**Fig. 5 Proposed mechanism and DFT calculation details. a** Proposed mechanism. **b** DFT calculations at the M06L-D3/6-311 + G(d,p)-SDD-SMD(DMA)//B3LYP-D3/6-31 G(d)-SMD(DMA) level of theory. Free energies are given in kcal/mol. **c** Optimized 3D structures of **TS1**, **TS2**, **TS3** and **TS4**.

using Tang and Xu's method[69]. Inspired by the precedents of Koh and others[15,67,68], we focused our attention on investigating the ligand effect of phosphine ligands (Fig. 4e). Both PPh₃ and PMe₃ altered the regioselectivity, and migratory β-alkylated amide (**6aa**) was isolated as the main product. After performing systematic screening of the influence of other reaction parameters, we determined that NiCl₂(PPh₃)₂ as the catalyst, (MeO)₃SiH as the hydride source, KF as the base, and in DMAc solvent provided a moderate 50% GC yield and 42% isolated yield with an excellent 19:1 β-alkylation selectivity (**6aa**). Representative substrates from the different dimensions of the directing group (**6ra**), chain length (**6qa**), and an internal alkene with considerable steric hindrance (**6ca**)

smoothly underwent the desired migratory hydroalkylation, demonstrating the generality and portability of the phosphine ligand-dominated β-alkylation selectivity (Fig. 4f).

**DFT calculations**. Based on previous literature and our experiments, we proposed two possible catalytic cycles for the proximal-selectivity hydroalkylation of unactivated alkenes (Fig. 5a)[31,43-45,66,68,70]. In the left catalytic cycle, Ni(I)X (**A**) was generated as an initial catalyst under the reductive conditions. Subsequently, Ni(I)X (**A**) was associated with an alkene to form a chelated Ni(I) species (**B**) which reacted with silane to

generate a crucial Ni(I)-H species (**C**). Then, the Ni(I)-H is inserted into the C–C double bond via a six- or seven-membered-ring mode, which controls alkylation position to offer Ni(I) species **D-1** or **D-2**. The resulting **D-1** abstracted halogen from an alkyl halide to form Ni(II) and an alkyl radical. The alkyl radical was trapped to give Ni(III) species (**F**), which underwent rapid reductive elimination to obtain the target product and regenerate catalyst **A**. Another possible catalytic cycle is shown on the right. Initially, Ni(I)X (**A**) reacted with an alkyl halide through halide abstraction to afford Ni(II) (**B'**) and an alkyl radical. Then, silane reacted with **B'** to generate the Ni(II)-H (**C'**), which chelated with an alkene to give Ni(II) species (**D'**). **D'** inserted into an alkene and captured the alkyl radical, followed by reductive elimination to obtain the target product (**D' → E' → F**). Preliminary DFT calculations indicate that the migration insertion of the Ni(II)-H bond into the terminal alkene group generating branched products via a six-membered-ring mode (**TS1** in Fig. 5b, barrier: 7.3 kcal/mol) is remarkably more facile than that of the seven-membered-ring-mode generating linear products (**TS2**, barrier: 16.2 kcal/mol). This result correlates with the preferential hydrogen atom incorporation on the terminal carbon atom observed experimentally. By contrast, the linear migration insertion process on Ni(I) system is excluded due to its high energy demands (**TS4**, barrier: 25.4 kcal/mol). However, Ni(I)-catalysed six-membered-ring migration insertion generating branched products is also possible under experimental conditions (**TS3**, barrier: 15.3 kcal/mol). Optimized 3D structures of **TS1**, **TS2**, **TS3** and **TS4** are shown in Fig. 5c. Totally, the energies for migration insertion steps in Ni(II)-H pathway are lower than that in Ni(I)-H pathway, but the possibility of Ni(I)-H pathway could not be ruled out at this stage.

In summary, we describe a nickel-catalysed proximal-selective hydroalkylation of unactivated alkenes to provide a retrosynthetic method for valuable β- or γ-branched alkyl carboxylic acid and β-, γ- or δ-branched alkyl amine building blocks. The sensible employment of aminoquinoline and picolinamide as suitable directing groups enables the hydroalkylation of both unactivated terminal and internal alkenes. A broad range of alkyl iodides and bromides with different functional groups can be installed with excellent regiocontrol to be available for site-selective late-stage functionalization of biorelevant molecules. Under modified reaction conditions using NiCl$_2$(PPh$_3$)$_2$ as the catalyst, migratory β-selective hydroalkylation can also be realized. Mechanistic studies and DFT calculations illuminate the ligand effects on regioselectivity and the detailed reaction mechanism. Further efforts on the catalytic asymmetric variant of this transformation are ongoing in our laboratory.

## Methods

### General procedure for proximal-selective hydroalkylation of unactivated alkenes.
In air, a 10 mL Schlenk tube equipped with a stir bar was charged with NiBr$_2$(diglyme) (0.02 mmol, 10 mol%), KF (0.6 mmol, 3.0 equiv), alkene (0.2 mmol, 1.0 equiv), and alkyl halide (0.4 mmol, 2.0 equiv) (0.5 equiv NaI was added when alkyl bromide was used). The Schlenk tube was evacuated and filled with argon (three cycles). To these solids, a solution of ethanolamine (0.024 mmol, 12 mol%) in anhydrous DMAc (1.0 mL) was added under an argon atmosphere. Then, DEMS (0.6 mmol, 3.0 equiv) (if the alkene or alkyl halide was liquid, it was also added at this time) was added under an argon atmosphere. The mixture was stirred at 25 °C or 40 °C for 12 h, diluted with H$_2$O followed by extraction with ethyl acetate, dried with anhydrous Na$_2$SO$_4$ and concentrated in vacuo. The residue was purified by column chromatography to produce the target product.

### General procedure for migratory hydroalkylation of unactivated alkenes.
In air, a 10 mL Schlenk tube equipped with a stir bar was charged with NiCl$_2$(PPh$_3$)$_2$ (0.02 mmol, 10 mol%), KF (0.6 mmol, 3.0 equiv), alkene (0.4 mmol, 2.0 equiv), and alkyl halide (0.2 mmol, 1.0 equiv). The Schlenk tube was evacuated and filled with argon (three cycles). To these solids, anhydrous DMAc (1.0 mL) was added under an argon atmosphere. Then, (MeO)$_3$SiH (0.6 mmol, 3.0 equiv) (if the alkene or

alkyl halide was liquid, it was also added at this time) was added under an argon atmosphere. The mixture was stirred at 100 °C for 12 h, diluted with H$_2$O followed by extraction with ethyl acetate, dried with anhydrous Na$_2$SO$_4$ and concentrated in vacuo. The residue was purified by column chromatography to produce the target product.

## Data availability
The authors declare that the data supporting the findings of this study are available within the article and Supplementary Information files or from the corresponding author upon request. The experimental procedures, computational results and characterization of all new compounds are provided in the Supplementary Information.

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

## Acknowledgements

This work received financial support from the National Natural Science Foundation of China (grant numbers 21732006, 51821006, and 51961135104 to Y.F. and grant number 21927814 to X.L.), the USTC Research Funds of the Double First-Class Initiative (grant number YD3530002002 to X.L.) and the CAS Collaborative Innovation Program of Hefei Science Center (grant number 2021HSC-CIP004 to Y.F. and X.L.). The numerical calculations in this paper have been done on the supercomputing system in the Supercomputing Center of the University of Science and Technology of China. X.L. wishes to acknowledge Zhenhua Gu (USTC) for insightful mechanistic discussion.

## Author contributions

X.L. and Y.F. directed the project. Y.F. directed and Y.-T.X. performed the DFT calculations. X.L. conceived the idea, designed the experiments and wrote the manuscript. X.-X.W. and Z.-L.Z. performed the synthetic experiments. All of the authors participated in the discussion and preparation of the manuscript.

## Competing interests

The authors declare no competing interests.
