## [Peer Review File · Nature Communications]

REVIEWER COMMENTS

Reviewer #1 (Remarks to the Author):

In this manuscript, Fu and co-workers describe a directed, proximal selective, nickel-catalyzed alkene hydroalkylation using silanes as hydride source and alkyl halides as alkyl electrophile, building on earlier work from the same group on NiH-mediated C–C couplings (Refs. 18, 19, 22, 40). A combination of 1,2-aminoalcohol ligand and 8-aminoquinoline amide directing group was key for the success of the transformation. Both terminal and internal alkenes were compatible substrates with minimal chain walking under the conditions shown in Table 2. The scope and functional group compatibility were demonstrated across >40 examples. A preliminary result demonstrating an asymmetric variant is reported (21%, 80% ee), which is notable despite being preliminary in nature. The authors also show that under modified reaction conditions with PPh₃ as ligand migratory hydroalkylation takes place to give β - (rather than γ -) functionalized products, echoing earlier findings by Koh (see below). The ability to turn on/off alkylnickel chain-walking by simple adjustments to reaction conditions is another notable finding in this study. DFT calculations were performed to shed light on the reaction mechanism.

In terms of relevant prior art, an important precedent comes from Koh and co-workers (Ref. 47), where regioconvergent (migratory) β -selective hydroalkylation of terminal and internal alkenyl amides was achieved using excess alkyl halide as both the alkyl and hydride source. The present work from Fu can be viewed as complementary in terms of the selectivity outcomes achieved. In this context, clearer description of Koh's work as well as other relevant precedents and more careful citations would improve the scholarship of this manuscript (see below). The Supporting Information contains all required elements.

Overall, there are some interesting aspects of this study, though my enthusiasm is tempered by some of the precedents described above which detract from the novelty of the work. Overall, I feel the manuscript may rise to the level of Nat. Commun., but some significant revisions will

be required prior to this point. These are described below.

1. On page 2, the linear functionalized products are not inherently “undesired” as the authors indicated. I recommend that this word be removed.

2. In the introduction, Refs. 19 and 58 are cited for Engle, but Ref. 19 appears to be by the corresponding author’s group.

3. On page 3, the authors write: “However, the remote hydrocarbonation process has never been realized.” This statement is probably not true without further qualification. One issue here (and elsewhere in the text) is that it’s unclear how the authors are defining ‘remote’ and ‘proximal’. The second issue is that almost irrespective of how they are defining it, there are examples that have in fact been reported, for instance, see γ - or δ -selective hydroarylation of similar alkenyl carbonyl compounds (Chem. Sci. 2018, 9, 6839; Nat. Commun. 2019, 5025; Ref. 66).

4. In the first paragraph of the introduction, the author state that NiH migratory insertion is potentially the rate-determining step, citing a general review article (Ref. 23). I would recommend that the authors either cite specific examples in the NiH literature to corroborate this statement, or to omit the “potentially the rate-determining step” claim, since it otherwise seems arbitrary.

5. In the introduction, prior work by Koh (Ref. 47) should be more clearly described in setting the stage for this study. Oddly, hydroamination precedents by Hong and Yu (Refs. 72 and 73) are given more attention than Ref. 47, despite being a clearly important precedent.

6. Regarding the data shown in Figure 3, with the addition of phosphine ligand found to promote chain walking, Hong (Nat. Commun. 2021, 5657) and Zhao (Nat. Commun. 2019, 5025) have documented similar phenomena in Ni-catalyzed migratory hydroamination and oxidative Heck couplings, respectively). Interesting, Hong found that chain walking stopped at the γ -position. It would be interesting for the authors to consider why the present hydroalkylation proceeds further to the β -position.

7. Page 6, line 3, the authors written “The Z/E configuration of the internal alkene would not

affect the coupling yield and regioselectivity.” However, I could not find an example where E and Z isomers for otherwise identical substrates were compared to corroborate this claim.

8. Throughout the figures, when r.r. values are quoted, it is not always immediately obvious what the structure of the minor regioisomer is. The authors should find a way to communicate this more clearly.

9. Some of the data presentation in Figure 3 could be improved. In panels A–C, substrates are sometime drawn out and sometimes not. Panel C is unnecessarily confusing because DG is not defined.

10. In Figure 3A, the yields are low (11 and 22%). What is the remaining material balance?

11. On Page 12, the authors show that TS1 and TS2 have what appear to be lower barriers than TS3 and Ts4. (It’s not immediately straightforward to compare since relative energies are quoted with respect to different zero-points.) However, no explanation is offered for why these low barriers can/should be disregarded (beyond the comment about not fitting with empirically observed selectivity).

12. Given the high barrier for TS4, did the authors consider a non-directed mechanism for this step?

13. In the DFT section, what is the rationale for only considering the OH form of the directing group. In the two related studies cited above (Refs. 72 and 73) the deprotonated form with Cs⁺ as cation was used.

14. There were some grammatical and typographical issues that can be corrected prior to publication.

- Abstract: “key to success was” should be “keys to success were”

- Page 2: “Lewis alkaline” should presumably be “Lewis basic”

- Page 3: “key to success was” should be “keys to success were”

- Page 4: change “susceptible to solvents” to “sensitive to solvent identity”

- Table 1, footnotes: ‘acac’ should be “acetylacetoate” (anionic form)

- Page 6: “and regioselectivity” should be “or regioselectivity”
- Page 6: delete “and furnished”
- Page 6: change “potential complete benzylic active” to “potentially competitive activated benzylic”
- Page 6: “aminoquinaldine” should presumably be “aminoquinoline” (two instances)
- Page 8: change ‘toleration’ to ‘tolerance’
- Table 2, Figures 2 and 3: “NMR” should be “¹H NMR”
- Page 12: “more feasible” should perhaps be “more facile”
- Page 12: “clarify the essence of the” should perhaps be “clarify the origin of”

Reviewer #2 (Remarks to the Author):

Lu, Fu and coworkers have developed a novel NiH-catalysed proximal-selective hydroalkylation of unactivated alkenes. By employing appropriate directing group and ligand, the regioselectivity of this chemistry could be well controlled. Importantly, data (f, figure 3) suggested the present reaction system could realize stop the chain working of NiH-species precisely at the β position. Furthermore, the great functional group tolerances lead to good ability of application for late-stage functionalization of biorelevant molecules. Moreover, the reaction pathway has been studied intensively by control experiments and DFT calculation. So this reviewer suggested the paper is suitable published on Nature Commun., after address the following questions.

1. A lot of valuable groups could compatible with this reaction, but all of them are primary carbon electrophiles. This reviewer suggested the secondary and tertiary electrophiles should be considered.

2. Both terminal and 1,2-disubstituted alkenes afforded great yield. How about 1,1-disubstituted alkene, trisubstituted alkene and tetrasubstituted alkene?
3. Interesting, with chiral ligand, the both enantioselectivity and regioselectivity could be control, but the yield was low (L6, e, Figure 3). How about the mass balance of this reaction? Is low reaction temperature helpful for avoiding side reaction or improve the ee of desired product?
4. To improve the utility for further transformation, the example about how to remove the directing group should be given.
5. The footnote of ¹¹B NMR at Supporting Page S45 was missed.
6. Some spectra of ¹³C NMR not clean, such as 4ab, 4ac, 4af, 4ao, 4ha, 4la and 7ba.

Reviewer #3 (Remarks to the Author):

This manuscript reports the NiH-catalysed proximal-selective hydroalkylation of unactivated alkenes to β - or γ -branched alkyl carboxylic acids and β -, γ - or δ -branched alkyl amines and the mechanistic studies were carried out. The author demonstrated that the employment of aminoquinoline and picolinamide as suitable directing groups enables the hydroalkylation of both unactivated terminal and internal alkenes. After detailed evaluation of this manuscript, I think it is suitable published on the Nature Communication before the major revision.

1. The manuscript contains some grammatical errors. In Page12, "By contrast, the energy demands for the two regioselective migration insertion step on Ni(II) system are differentiated by only 0.5 kcal/mol and is thus unlikely to explain the experimental results (Figure 4d)." step should be changed into steps.
2. It should be better if the Ni(II)-H mechanism catalytic cycle is rotated 90° clockwise so the comparison of the two cycles will be clearer.
3. Apart from the energy demands for the two regioselective migration insertion steps in the

Ni(II)-H mechanism are similar, the reason for the choice of the Ni(I)-H mechanism should be illustrated in detail as energies for migration insertion steps are lower than that in Ni(I)-H mechanism.

To sum up, this manuscript developed a nickel-catalysed proximal-selective hydroalkylation of unactivated alkenes. Through theoretical studies, authors provide sightful proofs for the reaction mechanism; therefore this manuscript could be accepted in Nature Communication after proposed corrections.

Point-by-point response to comments on NCOMMS-21-39983

We thank the reviewers for taking their valuable time to check the manuscript and for giving constructive suggestions to improve it. We have worked tirelessly to address every comment of the reviewers. A point-to-point response to the comments is as follows. The original reviewer comments are in blue.

In response to **Reviewer #1**:

In this manuscript, Fu and co-workers describe a directed, proximal selective, nickel-catalyzed alkene hydroalkylation using silanes as hydride source and alkyl halides as alkyl electrophile, building on earlier work from the same group on NiH-mediated C–C couplings (Refs. 18, 19, 22, 40). A combination of 1,2-aminoalcohol ligand and 8-aminoquinoline amide directing group was key for the success of the transformation. Both terminal and internal alkenes were compatible substrates with minimal chain walking under the conditions shown in Table 2. The scope and functional group compatibility were demonstrated across >40 examples. A preliminary result demonstrating an asymmetric variant is reported (21%, 80% ee), which is notable despite being preliminary in nature. The authors also show that under modified reaction conditions with PPh₃ as ligand migratory hydroalkylation takes place to give β- (rather than γ-) functionalized products, echoing earlier findings by Koh (see below). The ability to turn on/off alkylnickel chain-walking by simple adjustments to reaction conditions is another notable finding in this study. DFT calculations were performed to shed light on the reaction mechanism.

Response: We thank the reviewer for the positive comments, especially the positive comments on our efforts towards developing a ligand-controlled regiodivergent hydroalkylation of unactivated alkenes and an asymmetric variant. We added these descriptions into the introduction and conclusions as advantages of our reaction.

Using NiCl₂(PPh₃)₂ as the catalyst, migratory hydroalkylation takes place to provide β- (rather than γ-) branched products. A preliminary result with moderate enantioselectivity demonstrates that

an asymmetric variant could be realized.

Under modified reaction conditions using $\text{NiCl}_2(\text{PPh}_3)_2$ as the catalyst, migratory β -selective hydroalkylation could also be realized. Further efforts on the catalytic asymmetric variant of this transformation are ongoing in our lab.

In terms of relevant prior art, an important precedent comes from Koh and co-workers (Ref. 47), where regioconvergent (migratory) β -selective hydroalkylation of terminal and internal alkenyl amides was achieved using excess alkyl halide as both the alkyl and hydride source. The present work from Fu can be viewed as complementary in terms of the selectivity outcomes achieved. In this context, clearer description of Koh's work as well as other relevant precedents and more careful citations would improve the scholarship of this manuscript (see below). The Supporting Information contains all required elements.

Response: We have added an appropriate description of Koh's work and other relevant precedents. We believe without a doubt that previous works mentioned by this reviewer do not reduce the novelty of our work. On the contrary, they highlight the advantage and importance of our method. The more precise description and more careful citations improve the scholarship and readability of our manuscript. The significant revisions are shown below in the form of a point-to-point response to the comments.

Overall, there are some interesting aspects of this study, though my enthusiasm is tempered by some of the precedents described above which detract from the novelty of the work. Overall, I feel the manuscript may rise to the level of Nat. Commun., but some significant revisions will be required prior to this point. These are described below.

Response: We thank the reviewers for taking their precious time to check the manuscript and for giving invaluable advice to improve it. We have worked tirelessly to address every comment of the reviewers.

1. On page 2, the linear functionalized products are not inherently “undesired” as the authors indicated.

I recommend that this word be removed.

Response: The sentence has been revised to the following:

chain walking isomerization often occurs in internal alkenes and causes linear functionalized products

2. In the introduction, Refs. 19 and 58 are cited for Engle, but Ref. 19 appears to be by the corresponding author’s group.

Response: The erroneous citation has been revised.

3. On page 3, the authors write: “However, the remote hydrocarbonation process has never been realized.” This statement is probably not true without further qualification. One issue here (and elsewhere in the text) is that it’s unclear how the authors are defining ‘remote’ and ‘proximal’. The second issue is that almost irrespective of how they are defining it, there are examples that have in fact been reported, for instance, see γ - or δ -selective hydroarylation of similar alkenyl carbonyl compounds (*Chem. Sci.* 2018, 9, 6839; *Nat. Commun.* 2019, 5025; Ref. 66).

Response: We agree with the reviewer’s comments and made significant revisions.

The words ‘proximal-selective’ has been used in Hong’s description (*J. Am. Chem. Soc.* 2020, 142, 20470) as follows:

‘proximal-selective hydroamination methods that are applicable to a wide range of unactivated olefins have proven elusive’

Also see Engle’s description (*Angew. Chem. Int. Ed.* 2019, 58, 17068) as follows:

*‘across all of these examples, the phenyl group was reliably delivered to position **distal** to the directing group, and the Bpin group moiety was installed **proximal** to the directing group’*

In our revised Figure 1, we defined the ‘proximal-selective’ more clearly.

For a precise definition of ‘remote’, we referred to Hong’s description (*Nat. Commun.* **2021**, *12*, 5657) as follows:

‘selective functionalization of remote γ -C(sp³)-H bonds through controllable alkene transposition remains a significant unexplored challenge’ and ‘we were intrigued by the possibility of achieving remote γ -selective C(sp³)-H functionalization of unactivated alkenes’

(Screenshot from *Nat. Commun.* **2021**, *12*, 5657)

In the abstract, we have changed ‘the remote hydrocarbonation method applied to a broad range of unactivated alkenes remains challenging’ into ‘the branch-selective hydroalkylation method applied to a broad range of unactivated alkenes remains challenging’.

In the introduction, we have changed ‘despite the success of terminal-selective migratory hydrocarbonation of internal alkenes and α - or β -selective hydrocarbonation of conjugated alkenes or alkenes containing auxiliary groups, the remote hydrocarbonation method applied to a broad range of unactivated alkenes has proven elusive’ into ‘despite the success of terminal-selective migratory hydroalkylation of internal alkenes and α - or β -selective hydroalkylation of activated (conjugated) alkenes, the branch-selective hydroalkylation method applied to a broad range of unactivated alkenes remains challenging’.

We have changed ‘from the perspective of the previous literature, the hydroalkylation of internal alkenes with a remote position from functional groups remains challenging’ into ‘from the

perspective of the previous literature, hydroalkylation of internal alkenes on a remote position to functional units (rather than α - or β -alkylation) remains challenging’.

We deleted the sentence ‘however, the remote hydrocarbonation process has never been realized’.

Relevant elegant precedents on γ - or δ -selective hydroarylation of similar alkenyl carbonyl compounds (*Chem. Sci.* **2018**, *9*, 6839; *Nat. Commun.* **2019**, *10*, 5025; and *Angew. Chem. Int. Ed.* **2020**, *59*, 23306) have been cited as ref. 15, 23, and 25.

We thank the reviewers for giving invaluable advice to improve the manuscript. We would be happy to revise further if the reviewer has additional suggestions on the wordings.

4. In the first paragraph of the introduction, the author state that NiH migratory insertion is potentially the rate-determining step, citing a general review article (Ref. 23). I would recommend that the authors either cite specific examples in the NiH literature to corroborate this statement, or to omit the “potentially the rate-determining step” claim, since it otherwise seems arbitrary.

Response: The sentence has been revised to the following:

However, nickel-hydride migratory insertion is sensitive to alkene substrates' steric and electrical properties.

5. In the introduction, prior work by Koh (Ref. 47) should be more clearly described in setting the stage for this study. Oddly, hydroamination precedents by Hong and Yu (Refs. 72 and 73) are given more attention than Ref. 47, despite being a clearly important precedent.

Response: We thank the reviewer for giving invaluable advice to improve the readability of the introduction. We have added an appropriate description of Koh's work and made more careful citations on other relevant precedents on alkene hydroamination(amidation) and hydroarylation. Actually, the finding that PPh₃ as ligand enables the migratory β -selective (rather than γ -selective) hydroalkylation was inspired by Koh's work (*Nat. Commun.* **2020**, *11*, 5857, cited as ref. 68 in the revised manuscript).

We added sentences as following:

Recently, Koh and co-workers made progress in NiH-catalysed branch-selective hydroalkylation of unactivated alkenes using the aminoquinoline directing auxiliary. Alkene migratory hydroalkylation undergoes smoothly to access β -alkylated amides with excellent regioselectivity.

We added the reaction equation of Koh's work into Figure 1:

6. Regarding the data shown in Figure 3, with the addition of phosphine ligand found to promote chain walking, Hong (Nat. Commun. 2021, 5657) and Zhao (Nat. Commun. 2019, 5025) have documented similar phenomena in Ni-catalyzed migratory hydroamination and oxidative Heck couplings, respectively). Interesting, Hong found that chain walking stopped at the γ -position. It would be interesting for the authors to consider why the present hydroalkylation proceeds further to the β -position.

Response: These precedents inspired us to investigate the ligand effect of phosphine ligands. Both of these precedents indicate the importance of the formation of stable **five-membered nickelacycle** or **six-membered metallacycle intermediate**. The stable metallacycle intermediate causes a strongly thermodynamically favoured termination step and is the driving force to terminate the selective alkene migration.

We added one sentence as follows:

Inspired by the precedents of Koh and others (Nat. Commun. 2020, 11, 5857 as ref. 68; Nat. Commun. 2021, 12, 5657 as ref. 67; and Nat. Commun. 2019, 10, 5025 as ref. 15), we focused our attention on investigating the ligand effect of phosphine ligands.

In Koh's work (Nat. Commun. 2020, 11, 5857), the authors postulated that '*the organonickel intermediate **ix** generated from nickel-hydride addition isomerized to the relatively more stabilized **five-membered nickelacycle x** through consecutive β -H elimination/olefin insertion steps*', and then '*x could be converted to **18** with net remote alkylation at the β position*'.

In Hong's work (*Nat. Commun.* **2021**, *12*, 5657), the authors suggested a AQ-chelated six-membered metallacycle intermediate was responsible for γ -selective amination.

'Controlled by the stability of the AQ-chelated six-membered complex, selective cross-coupling with an aminating reagent readily proceeds through oxidative addition and reductive elimination processes to afford the desired γ -aminated product'

In Zhao's work (*Nat. Commun.* **2019**, *10*, 5025), the authors described the key steps and intermediates as follows:

'Six-membered nickelacycle C1-1 directly loses a second PMe_3 ligand upon coordination of the quinoline group to form a more stable species C2 (a six-membered nickelacycle).'

'Subsequent carbometalation across the $C=C$ bond via TS2B leads to formation of linear intermediate C'2 (also a six-membered nickelacycle)'

In our independent research on alkene migratory hydroalkylation, we found that both PPh_3 ligand and 100 °C high reaction temperature were responsible for β -selective hydroalkylation. With the addition of PPh_3 , a mixture of β - and γ -selective products was obtained. The ratio of β -/ γ -selective products increases concurrently with the increase of reaction temperature (entries 3, 4). We could obtain β -selective alkylated products with high regioselectivity and moderate coupling yield at 100 °C. Further studies on the development of an efficient method to access γ -selective with high regioselectivity and coupling efficiency based on ligand design are in progress. Actually, we have found that the reaction shows an increasing tendency on γ -selectivity with the addition of external ethanolamine (MEA) ligand to the conditions for migratory hydroalkylation (entry 5).

entry	variation from standard conditions C	yield/% ^a	r.r. ^a
1	none	41 (35 ^b)	14:1
2	15 °C	33	1:2.7
3	60 °C	23	1.6:1
4	external 12% ethanolamine	49	1:1.5

^aYields and regioisomeric ratios were determined by GC analysis with triphenylmethane as an internal standard. Total yield for the mixture of all regioisomers. r.r. refers to the regioisomeric ratio, that of the β -selective product to the sum of all other isomers. In most cases, a mixture of β - and γ -selective products was obtained; we could hardly observe other isomers. ^bIsolated yield in parentheses.

With related literature and our experimental data, it became clear that (1) phosphine ligand promotes the chain-walking process and (2) both **five- and six-membered nickelacycle** could be stable intermediates in alkene migratory hydrofunctionalization and deliver the corresponding β - and γ -functionalized products. Under our reaction condition with a PPh₃ ligand and a high reaction temperature (100 °C), a thermodynamically stable **five-membered nickelacycle** formed with high selectivity and was responsible for β -selective hydroalkylation.

7. Page 6, line 3, the authors written “The Z/E configuration of the internal alkene would not affect the coupling yield and regioselectivity.” However, I could not find an example where E and Z isomers for otherwise identical substrates were compared to corroborate this claim.

Response: We agree with the reviewer’s comment. We synthesized **E-1b** ($Z/E < 1:20$) and **Z-1b** ($Z/E > 20:1$) to conduct control experiment. Both **Z-** and **E-** configuration starting materials performed well in the transformation. Therefore, we concluded that the Z/E configuration of the internal alkene would not affect the coupling yield or regioselectivity.

8. Throughout the figures, when r.r. values are quoted, it is not always immediately obvious what the structure of the minor regioisomer is. The authors should find a way to communicate this more clearly.

Response: We have modified all tables and figures to show the minor regioisomer more clearly.

(1) For terminal alkene **1a** under standard conditions, we obtained γ -selective hydroalkylation product **4aa** as the major regioisomer, and δ -hydroalkylation product **4aa'** as the minor regioisomer; other regioisomers were not observed (lower than the detection limit of gas chromatography).

GC spectra of **4aa** under standard conditions

(2) For internal alkene **1d** under standard conditions, the outcome was similar to terminal alkene **1a** under standard conditions.

GC spectra of **4da**

(3) For terminal alkene **1a** under asymmetric catalytic conditions, the outcome was similar to terminal alkene **1a** under standard conditions.

GC spectra of **4aa** under asymmetric catalytic conditions

(4) For terminal alkene **1a** under migratory β -selective conditions, we obtained β -selective hydroalkylation product **6aa** as the major regioisomer, and γ -hydroalkylation product **6aa'** as the minor regioisomer; other regioisomers were not observed (lower than the detection limit of gas chromatography).

GC spectra of **6aa** under migratory β -selective conditions

We added the necessary notes as follows:

r.r. refers to the regioisomeric ratio, that of the major product to the sum of all other isomers.

For both racemic and asymmetric alkene *ipso*-hydroalkylation, proximal-selective hydroalkylation product was obtained as the major regioisomer, and distal-selective hydroalkylation product as the minor regioisomer; other regioisomers could hardly be detected.

For alkene migratory β -selective hydroalkylation, β -selective hydroalkylation product was obtained as the major regioisomer, and γ -selective hydroalkylation product as the minor regioisomer; other regioisomers could hardly be detected.

9. Some of the data presentation in Figure 3 could be improved. In panels A–C, substrates are sometime drawn out and sometimes not. Panel C is unnecessarily confusing because DG is not defined.

Response: We have modified Figure 3 according to the reviewer's suggestion. This figure now appears as:

10. In Figure 3A, the yields are low (11 and 22%). What is the remaining material balance?

Response: Our original intention was to show the critical outcomes to support the radical nature of this hydroalkylation reaction. Coupling product **4ay** originating from the ring-opening of the cyclopropylmethyl radical and ring-cyclized product **4az** originated from the cyclization of the 5-hexenyl radical support the conclusion.

Original figure 3A imitated Hu's drawing mode (see *Nat. Chem.* **2021**, 13, 270). Hu's description and conclusion as follows:

'The product (13) originated from cyclization of the 5-hexenyl radical and was obtained in 11% yield and 91% e.e.. The coupling product that originated from the uncyclized 5-hexenyl radical was also detected but it was difficult to purify and isolate. These data support the intermediacy of alkyl radicals.'

(Screenshot from *Nat. Chem.* **2021**, 13, 270)

According to the reviewer's comment, we revised the drawing mode of figure 3A and detailed information was added to SI.

For the reaction using (bromomethyl)cyclopropane (**3y**) as an electrophile, we obtained **4ay** and other ring-opened products (**4aya**, **4ayb**, and **4ayc**) with the migration or hydrogenation of alkenyl double bond.

GC spectra of radical clock experiment using (bromomethyl)cyclopropane

¹H NMR spectra of radical clock experiment using (bromomethyl)cyclopropane

For the reaction using 6-iodohex-1-ene (**2z**) as an electrophile, we obtained ring-cyclized product **4az** and other uncyclized products (**4aza**, **4azb**, and **4azc**) with the migration or hydrogenation of alkenyl double bond.

other (by)products

GC spectra of radical clock experiment using 6-iodohex-1-ene

¹H NMR spectra of radical clock experiment using 6-iodohex-1-ene

11. On Page 12, the authors show that TS1 and TS2 have what appear to be lower barriers than TS3 and Ts4. (It's not immediately straightforward to compare since relative energies are quoted with respect to different zero-points.) However, no explanation is offered for why these low barriers can/should be disregarded (beyond the comment about not fitting with empirically observed selectivity).

Response: We agree to the comment and thank the reviewer for his/her concern. According to question 11 and question 13, we have rebuilt the energy profiles in our revised manuscript and used the energy of **Int1** as the only zero-point in our work.

In short, Ni(II)-H involved mechanistic pathway is more plausible than Ni(I)-H pathway, but the possibility of Ni(I)-H pathway could not be ruled out at this stage.

12. Given the high barrier for TS4, did the authors consider a non-directed mechanism for this step?

Response: We examined the energy demand for the proposed pathway involving a non-directed mechanism. The calculation results indicate that the energy barrier of the proposed non-directed transition state (**TS4-1**) is higher than that of the directed transition state **TS4** by 13.6 kcal/mol, which is unlikely to occur under experimental conditions (25 °C). These results indicate the importance of the directing group. The aminoquinoline amide directing group plays a major role to lower the activation barrier for the migratory insertion through coordination with the metal centre.

Energy profiles of Ni(I)-catalysed migratory insertion

(the Gibbs free energies are given in kcal/mol)

13. In the DFT section, what is the rationale for only considering the OH form of the directing group. In the two related studies cited above (Refs. 72 and 73) the deprotonated form with Cs⁺ as cation was used.

Response: We thank the reviewer for this crucial comment on our DFT calculation section. We made profound consideration on the possibility of a deprotonated form with alkali metal ions (K⁺ in our work).

DFT calculations considering -OK form (rather than -OH form) of the directing group were performed, and results are provided (see below). Ni(II)-catalysed migratory insertion is more likely to occur, and branched products are the main product (only needs to overcome an energy barrier of 7.3 kcal/mol). Ni(II)-catalysed migratory insertion step is both regio-determining and rate-determining step. In addition, Ni(I)-catalysed branched migratory insertion is also possible under experimental temperature (25 °C). Ni(I)-catalysed linear migratory insertion is not likely to occur (with an energy barrier of 25.4 kcal/mol). Totally, the energies for migration insertion steps in Ni(II)-H pathway are lower than that in Ni(I)-H pathway, but the possibility of Ni(I)-H pathway could not be ruled out at this stage.

The new calculation results are provided in Figure 5 of the revised manuscript.

Preliminary DFT calculations indicate that the migration insertion of the Ni(II)-H bond into the terminal alkene group generating branched products via a six-membered-ring mode (**TS1** in Figure 5b, barrier: 7.3 kcal/mol) is remarkably more feasible than that of the seven-membered-ring-mode generating linear products (**TS2**, barrier: 16.2 kcal/mol). This result correlates with the preferential hydrogenation on the terminal carbon atom observed experimentally. By contrast, the linear migration insertion process on Ni(I) system is excluded due to its high energy demands (**TS4**, barrier: 25.4 kcal/mol). However, Ni(I)-catalysed six-membered-ring migration insertion generating branched products is also possible under experimental conditions (**TS3**, barrier: 15.3 kcal/mol). Totally, the energies for migration insertion steps in Ni(II)-H pathway are lower than that in Ni(I)-H pathway, but the possibility of Ni(I)-H pathway could not be ruled out at this stage.

Energy profiles of Ni-catalysed mechanism

(the Gibbs free energies are given in kcal/mol)

14. *There were some grammatical and typographical issues that can be corrected prior to publication.*

- *Abstract: “key to success was” should be “keys to success were”*
- *Page 2: “Lewis alkaline” should presumably be “Lewis basic”*
- *Page 3: “key to success was” should be “keys to success were”*
- *Page 4: change “susceptible to solvents” to “sensitive to solvent identity”*
- *Table 1, footnotes: ‘acac’ should be “acetylacetoate” (anionic form)*
- *Page 6: “and regioselectivity” should be “or regioselectivity”*
- *Page 6: delete “and furnished”*
- *Page 6: change “potential completive benzylic active” to “potentially competitive activated benzylic”*
- *Page 6: “aminoquinaldine” should presumably be “aminoquinoline” (two instances)*
- *Page 8: change ‘toleration’ to ‘tolerance’*
- *Table 2, Figures 2 and 3: “NMR” should be “¹H NMR”*
- *Page 12: “more feasible” should perhaps be “more facile”*
- *Page 12: “clarify the essence of the” should perhaps be “clarify the origin of”*

Response: We thank the reviewer for the thorough examination of the manuscript. We have corrected mentioned grammatical issues and fixed several typos.

In response to **Reviewer #2:**

Lu, Fu and coworkers have developed a novel NiH-catalysed proximal-selective hydroalkylation of unactivated alkenes. By employing appropriate directing group and ligand, the regioselectivity of this chemistry could be well controlled. Importantly, data (f, figure 3) suggested the present reaction system

could realize stop the chain working of NiH-species precisely at the β position. Furthermore, the great functional group tolerances lead to good ability of application for late-stage functionalization of biorelevant molecules. Moreover, the reaction pathway has been studied intensively by control experiments and DFT calculation. So this reviewer suggested the paper is suitable published on *Nature Commun.*, after address the following questions.

Response: We thank the reviewer for the positive comments and the support of the publication of our manuscript on *Nature Communications* after appropriate revisions.

1. A lot of valuable groups could compatible with this reaction, but all of them are primary carbon electrophiles. This reviewer suggested the secondary and tertiary electrophiles should be considered.

Response: We examined several representative secondary and tertiary electrophiles.

For the reaction using iodocyclohexane (**2B**) as an electrophile, we obtained the hydroalkylation product in 23% yield (2.4:1 r.r.) and the alkene protonation product in 38% yield. In addition, the starting material **1a** was recovered in a 7% yield. We examined 3 different bases and 2 different silanes to improve the coupling efficiency and regioselectivity. The results are shown below.

entry	variation from standard conditions A	yield/% ^a	r.r. ^a
1	none	26 (23 ^b)	2.4:1
2	CsF instead of KF	11	1:2.1
3	K ₃ PO ₄ (H ₂ O) instead of KF	13	1:1.5
4	NaF instead of KF	11	3.9:1
5	(EtO) ₃ SiH instead of DEMS	27	1.2:1
6	(MeO) ₃ SiH instead of DEMS	40 (35 ^b)	1.9:1
7	80 °C	32	1.4:1

^aYields and regioisomeric ratios were determined by GC analysis with triphenylmethane as an

internal standard. Total yield for the mixture of all regioisomers. r.r. refers to the regioisomeric ratio, that of the major product to the sum of all other isomers. In most cases, a mixture of γ - and δ -selective products was obtained; we could hardly observe other isomers. ^bIsolated yield in parentheses.

For the reaction using tertiary electrophiles, such as 2-iodo-2-methylpropane (**2C**), 2-bromo-2-methylpropane (**3C**), and 1-bromoadamantane (**3D**), the desired hydroalkylation products were not observed. The alkene protonation product was observed as the main by-product with a large amount of starting material **1a** recovered.

We conducted the reaction of activated secondary electrophile (ethyl 2-bromobutanoate, **3E**) under standard conditions. The linear-selective hydroalkylation product **4aE'** was obtained in 20% yield and the alkene protonation product in 50% yield.

Therefore, this reaction might be sensitive to the steric hindrance of alkyl halides.

2. Both terminal and 1,2-disubstituted alkenes afforded great yield. How about 1,1-disubstituted alkene, trisubstituted alkene and tetrasubstituted alkene?

Response: We examined several representative multi-substituted alkenes.

In the example of trisubstituted alkene **4ga**, a secondary-primary C(sp³)-C(sp³) bond formed with a moderate yield. However, the expectation of constructing a tertiary-primary C(sp³)-C(sp³) bond failed in the examination of 1,1-disubstituted alkene and tetrasubstituted alkene.

For the reaction using 1,1-disubstituted alkene 4-methyl-*N*-(quinolin-8-yl)pent-4-enamide (**1s**), we obtained a small amount (< 10%) of linear-selective product **4sa'**. But γ -functionalized product **4sa** were not observed. In addition, the alkene protonation product was observed as the main by-product in a 68% yield.

The results for tetrasubstituted alkene 4,5-dimethyl-*N*-(quinolin-8-yl)hex-4-enamide (**1t**) were similar to 1,1-disubstituted alkene 4-methyl-*N*-(quinolin-8-yl)pent-4-enamide.

Combined with the results in question 2 and question 3, we added more description as the reaction limitations in the main text as follows:

We examined several secondary and tertiary electrophiles (see Supplementary Table 6 and Supplementary Figure 18 for more details). For the reaction using iodocyclohexane as an electrophile, the hydroalkylation product was delivered in a 23% yield (2.4:1 r.r.) with the alkene protonation byproduct in a 38% yield; slight modification using $(\text{MeO})_3\text{SiH}$ as hydride source gave out a 35% total yield (1.9:1 r.r.). For the reaction using tertiary electrophiles, such as 2-iodo-2-methylpropane (**2C**), 2-bromo-2-methylpropane (**3C**), and 1-bromoadamantane (**3D**), the desired hydroalkylation products were not observed. In the example of trisubstituted alkene **4ga**, a secondary-primary $\text{C}(\text{sp}^3)\text{-C}(\text{sp}^3)$ bond formed with a moderate yield. However, the expectation of constructing a tertiary-primary $\text{C}(\text{sp}^3)\text{-C}(\text{sp}^3)$ bond failed in the examination of 1,1-disubstituted alkene (**1s**) and tetrasubstituted alkene (**1t**) (see Supplementary Figure 19 for more details). Therefore, this reaction might be sensitive to the steric hindrance of both alkyl halides and alkene substrates.

3. Interesting, with chiral ligand, the both enantioselectivity and regioselectivity could be control, but

the yield was low (L6, e, Figure 3). How about the mass balance of this reaction? Is low reaction temperature helpful for avoiding side reaction or improve the ee of desired product?

Response: This comment contains two parts.

(1) *How about the mass balance of this reaction?*

Under the asymmetric catalytic conditions, the desired product was obtained in a 21% yield with 66% starting material **1a** recovered.

(2) *Is low reaction temperature helpful for avoiding side reaction or improve the ee of desired product?*

The main trend is that low reaction temperature helps to improve the e.e. value of the desired product. The results with lower temperature 5 °C and higher temperature 40 °C and 60 °C are shown (see below).

entry	variation from standard conditions B	yield/% ^a	r.r. ^a	e.e./% ^b
1	none	21	20:1	80
2	5 °C	19	20:1	70
3	40 °C	64	20:1	72
4	60 °C	61	20:1	58

^aYields and regioisomeric ratios were determined by GC analysis with triphenylmethane as an internal standard. Total yield for the mixture of all regioisomers. r.r. refers to the regioisomeric ratio, that of the major product to the sum of all other isomers. In all most cases, a mixture of γ - and δ -selective products was obtained; we could hardly observe other isomers. ^bThe e.e. value was determined by high-performance liquid chromatography (HPLC).

(3) We have included the results for asymmetric hydroalkylation containing 3 different nickel catalysts, 8 different ligands, in 7 different solvents, using 4 different silanes and 5 different

bases and at 4 different temperatures. These results are shown in Supplementary Tables 3–4.

4. To improve the utility for further transformation, the example about how to remove the directing group should be given.

Response: We conducted a scale-up reaction to obtain the alkene hydroalkylation product **4aa** in gram-scale in Figure 3b. Then, we transformed product **4aa** into methyl carboxylate **5a** and alkyl carboxylic acid **5b**. The directing group 8-aminoquinoline was proven to be readily recyclable.

In a scale-up reaction (Figure 3b), we achieved alkene hydroalkylation product **4aa** with a satisfactory 78% isolated yield and 12:1 regioselectivity. Under mild nickel-catalysed conditions, the directing amide group was transformed into a methyl ester group to deliver the desired methyl carboxylate **5a** in a 70% isolated yield. After hydrolysis of **4aa** in hydrochloric acid solution, alkyl carboxylic acid **5b** was obtained in a 73% isolated yield with 96% aminoquinoline recovered. The gram-scale reaction and successful removal of the directing group highlight the practicality of this nickel-catalysed alkyl-alkyl coupling reaction.

5. The footnote of 11B NMR at Supporting Page S45 was missed.

Response: We thank the reviewer for the thorough examination of the SI. We have revised this typographical issue.

6. Some spectra of ^{13}C NMR not clean, such as 4ab, 4ac, 4af, 4ao, 4ha, 4la and 7ba.

Response: We thank the reviewer for the thorough examination of the SI.

The purity of some ^{13}C NMR spectrum is barely satisfactory. We thank the reviewer for pointing out this issue. This ^{13}C NMR spectrum issue is not caused by the formation of alkene protonation products, but is due to the regioselective nature for the reaction. The polarities of the desired product and the other regioisomers are very close. Therefore, it can not be effectively separated by column chromatography. As a result, the purity of the ^{13}C NMR spectrum, especially the high field in ^{13}C NMR spectrum, is barely satisfactory.

For compound **7ba**, we synthesised the corresponding regioisomer **7ba'** for comparison. It is definite that the unsatisfactory purity of the ^{13}C NMR spectrum, especially the high field in ^{13}C NMR spectrum, is caused by the inseparable regioisomers.

Original ^{13}C NMR spectra for **7ba**

New ¹³C NMR spectra for **7ba**

¹H NMR spectra for regioisomer **7ba'**

¹³C NMR spectra for regioisomer **7ba'**

For compound **4ab**, we improved the purity of ¹³C NMR spectrum after more careful column chromatography separation.

Original ¹³C NMR spectra for **4ab**

New ¹³C NMR spectra for **4ab**

For compound **4ao**, **4ha**, and **4la**, we used the corresponding alkyl bromides instead of alkyl iodides, the regioselectivities were improved at different degrees. We improved the purity of ¹³C NMR spectrum for these compounds after more careful column chromatography separation.

Original ^{13}C NMR spectra for **4ao**

New ^{13}C NMR spectra for **4ao**

Original ^{13}C NMR spectra for **4ha**

New ^{13}C NMR spectra for **4ha**

Original ^{13}C NMR spectra for **4la**

New ^{13}C NMR spectra for **4la**

For compound **4ac** and **4af**, the purity of ^{13}C NMR spectrum were mostly unchanging even after

more careful column chromatography separation.

Original ^{13}C NMR spectra for **4af**

New ^{13}C NMR spectra for **4af**

In response to **Reviewer #3**:

This manuscript reports the NiH-catalysed proximal-selective hydroalkylation of unactivated alkenes to β - or γ -branched alkyl carboxylic acids and β -, γ - or δ -branched alkyl amines and the mechanistic studies were carried out. The author demonstrated that the employment of aminoquinoline and picolinamide as suitable directing groups enables the hydroalkylation of both unactivated terminal and internal alkenes. After detailed evaluation of this manuscript, I think it is suitable published on the Nature Communication before the major revision.

Response: We thank the reviewer for the support of the publication of our manuscript on *Nature Communications* after appropriate revisions.

1. The manuscript contains some grammatical errors. In Page12, "By contrast, the energy demands for the two regioselective migration insertion step on Ni(II) system are differentiated by only 0.5 kcal/mol and is thus unlikely to explain the experimental results (Figure 4d)." step should be changed into steps.

Response: We have fixed several typos.

2. It should be better if the Ni(II)-H mechanism catalytic cycle is rotated 90° clockwise so the comparison of the two cycles will be clearer.

Response: We adjusted the drawing of Ni(II)-H mechanism catalytic cycle. This figure now appears as follows:

3. Apart from the energy demands for the two regioselective migration insertion steps in the Ni(II)-H mechanism are similar, the reason for the choice of the Ni(I)-H mechanism should be illustrated in detail as energies for migration insertion steps are lower than that in Ni(I)-H mechanism.

Response: We thank the reviewer for this crucial comment. This suggestion is similar to questions 11 and 13 from reviewer #1. We made profound consideration on the possibility of a deprotonated form with alkali metal ions (K^+ in our work) and rebuilt the energy profiles in our revised manuscript. Totally, the energies for migration insertion steps in Ni(II)-H pathway are lower than that in Ni(I)-H pathway, but the possibility of Ni(I)-H pathway could not be ruled out at this stage.

The new calculation results are provided in Figure 5 of the revised manuscript.

To sum up, this manuscript developed a nickel-catalysed proximal-selective hydroalkylation of unactivated alkenes. Through theoretical studies, authors provide sightful proofs for the reaction mechanism; therefore this manuscript could be accepted in Nature Communication after proposed corrections.

Response: We thank the reviewer for the support of the publication of our manuscript on *Nature Communications* after proposed corrections.

REVIEWERS' COMMENTS

Reviewer #1 (Remarks to the Author):

The effort that has gone into this revision is substantial. The authors took the feedback from the reviewers seriously and made a good faith effort to improve the manuscript. In the revised version of the manuscript, Fu and co-workers reworded and reorganized the introduction part, using much more precise language. Key precedents are now highlighted and discussed in the appropriate context. Additional details are provided in the optimization table and in the experimental mechanistic section. A lot of new DFT data is presented in response to the suggestion to consider the deprotonated form of the directing group, which appears to be a lower energy pathway than the originally presented pathway. There are a few minor points that can be improved, but overall, the manuscript is essentially ready for publication as-is.

1. In the DFT section, the authors provided some data that was not discussed in the main text or in the Supporting Information, namely the alkyl bromide incorporation steps and the structural analysis of the four transition states (bond length, bifacial angles) in Fig 5c. Some commentary in the main text or the SI would help the reader understand what conclusions should be drawn from this data.

2. Also in regards to the DFT section, five-coordinate Ni(I) species are not commonly proposed as intermediates or transition states. Though this pathway is by no means impossible, it is notable that the Ni–N bond is long (3.39 Å or 3.60 Å). Commentary here would be welcome. Have the authors considered the possibility that the ligand is dissociated at this stage?

3. Overall, the writing is reasonably good and easy to follow. There were nevertheless a few minor grammatical and stylistic issues that can be improved:

- Page 2, line 46: change 'electrical' to 'electronic'
- Page 2, line 54: "terminal-selective" would perhaps be clearer as "linear-selective"
- Page 3, Line 75: 'demonstrate' should be 'demonstrates'
- Figure 1, the authors are encouraged to use a different color for DG, as the green color doesn't have much contrast from the blue, particularly when printed out.
- Figure 1: "ligand-regulatable" (with hyphen)
- Page 7, Line 132: change "a more vital directing ability over" to "stronger directing ability compared to"
- Page 7, Line 134: move 'selectively' before 'converted'

- Page 7, Line 142: delete comma before “and propylidene”
- Page 7, Line 152: insert ‘along’ before “with the”
- Page 7, Line 157: change “expectation of constructing” to “attempted construction of”
- Page 11, line 196: change ‘assumed’ to ‘expected’
- Page 12, line 217: ‘reversed’ would be clearer as ‘altered’. (I would assume ‘reversed’ regioselectivity would refer to the linear product.)
- Page 14, Line 248: change “in the right” to “on the right”
- Page 14, Line 256: change ‘hydrogenation’ to “hydrogen atom incorporation”

Reviewer #2 (Remarks to the Author):

This work has been carefully improved by the authors. Importantly, all of the answers and suggestions from reviewers have been well responded point by point. This reviewer suggested that it could be publish on nature communication as a high level article.

Reviewer #3 (Remarks to the Author):

< In remarks to the editorial office, the reviewer recommended publication as-is. >

A point-by-point response to reviewers' comments on NCOMMS-21-39983A

The reviewers addressed a few minor points. We made revisions with all three reviewers' help. We highly appreciate the reviewers for taking their valuable time to check our manuscript and giving constructive suggestions.

A point-to-point response to the comments is as follows. The original reviewer comments are in blue.

In response to **Reviewer #1**:

The effort that has gone into this revision is substantial. The authors took the feedback from the reviewers seriously and made a good faith effort to improve the manuscript. In the revised version of the manuscript, Fu and co-workers reworded and reorganized the introduction part, using much more precise language. Key precedents are now highlighted and discussed in the appropriate context. Additional details are provided in the optimization table and in the experimental mechanistic section. A lot of new DFT data is presented in response to the suggestion to consider the deprotonated form of the directing group, which appears to be a lower energy pathway than the originally presented pathway. There are a few minor points that can be improved, but overall, the manuscript is essentially ready for publication as-is.

Response: We thank the reviewer for the positive comments and the support of the publication of our manuscript on *Nature Communications*. We highly appreciate the reviewer for taking her/his precious time to check the manuscript and giving invaluable advice to improve it.

1. In the DFT section, the authors provided some data that was not discussed in the main text or in the Supporting Information, namely the alkyl bromide incorporation steps and the structural analysis of the four transition states (bond length, bifacial angles) in Fig 5c. Some commentary in the main text or the SI would help the reader understand what conclusions should be drawn from this data.

Response: We have added an appropriate description of the alkyl bromide incorporation steps and the structural analysis of the four transition states.

The following description and discussion have been added to the Supplementary Information.

In the proposed Ni(I)-catalyzed branched migratory insertion, nickel(I)-alkyl species **Int5** abstracts halogen from an alkyl halide to form nickel(II)-alkyl species **Int2** and an alkyl radical, passing over the transition state **TS5** with an energy barrier of 5.1 kcal/mol.

In the reductive elimination step, the alkyl radical is trapped by nickel(II)-alkyl species **Int2-1** to give Ni(III) species **TS6** and undergo rapid reductive elimination to complete the catalytic cycle.

3D structures of four transition states, **TS1**, **TS2**, **TS3** and **TS4**, are provided to explain the reaction regioselectivity clearly. The bifacial angle of C₁-C₂-N₁-C₃ in **TS2** appears to be -84.1°. Compared to **TS1** (the bifacial angle of C₁-C₂-N₁-C₃ is 164.3°), the twisted seven-membered ring in **TS2** breaks the conjugate structure of the alkene ligand. The bifacial angle in **TS2** almost changes from plane to horizontal, which is a state with significant steric hindrance. Ni₁-Ni in **TS2** has a much longer bond length of 2.63 Å compared to 2.00 Å in **TS1**, indicating that this distorted structure weakens the Ni₁-Ni coordination. Therefore, **TS2** has higher relative energy than **TS1**.

N₂-Ni bonds in **TS3** and **TS4** are much longer than the sum of van der Waals radii of nickel and nitrogen (3.39 Å > 3.25 Å, 3.60 Å > 3.25 Å), suggesting weak coordinations close to being ignored. Moreover, a similar twisted structure of **TS4** also causes the destabilization of the Ni₁-Ni bond (2.08 Å in **TS4** versus 1.94 Å in **TS3**), making it less stable than **TS3**.

2. Also in regards to the DFT section, five-coordinate Ni(I) species are not commonly proposed as intermediates or transition states. Though this pathway is by no means impossible, it is notable that the Ni–N bond is long (3.39 Å or 3.60 Å). Commentary here would be welcome. Have the authors considered the possibility that the ligand is dissociated at this stage?

Response: We agree to the comment and thank the reviewer for her/his concern. Actually, two Ni(I) species, **TS3** and **TS4**, seem to be more like four-coordinate (with square planar structures) after the structure optimization. The Ni–NH₂CH₂CH₂OH bonds in these two transition states are long. The interactions between Ni(I) species and the ethanolamine part can be ignored. We put the Ni(I) catalyst and the ethanolamine parts in one input file to reduce systematic error. To avoid misunderstanding, we have modified the structures of **TS3** and **TS4** in Figure 5b by changing Ni–NH₂CH₂CH₂OH bonds into dash lines (Ni···NH₂CH₂CH₂OH).

3. Overall, the writing is reasonably good and easy to follow. There were nevertheless a few minor grammatical and stylistic issues that can be improved:

- Page 2, line 46: change ‘electrical’ to ‘electronic’
- Page 2, line 54: “terminal-selective” would perhaps be clearer as “linear-selective”
- Page 3, Line 75: ‘demonstrate’ should be ‘demonstrates’
- Figure 1, the authors are encouraged to use a different color for DG, as the green color doesn’t have much contrast from the blue, particularly when printed out.
- Figure 1: “ligand-regulatable” (with hyphen)
- Page 7, Line 132: change “a more vital directing ability over” to “stronger directing ability compared to”
- Page 7, Line 134: move ‘selectively’ before ‘converted’
- Page 7, Line 142: delete comma before “and propylidene”
- Page 7, Line 152: insert ‘along’ before “with the”

- Page 7, Line 157: change “expectation of constructing” to “attempted construction of”
- Page 11, line 196: change ‘assumed’ to ‘expected’
- Page 12, line 217: ‘reversed’ would be clearer as ‘altered’. (I would assume ‘reversed’ regioselectivity would refer to the linear product.)
- Page 14, Line 248: change “in the right” to “on the right”
- Page 14, Line 256: change ‘hydrogenation’ to “hydrogen atom incorporation”

Response: We thank the reviewer for the thorough examination of the manuscript. We have corrected mentioned grammatical and stylistic issues.

In response to **Reviewer #2:**

This work has been carefully improved by the authors. Importantly, all of the answers and suggestions from reviewers have been well responded point by point. This reviewer suggested that it could be publish on nature communication as a high level article.

Response: We thank the reviewer for the support of the publication of our manuscript on *Nature Communications*.

In response to **Reviewer #3:**

< In remarks to the editorial office, the reviewer recommended publication as-is. >

Response: We thank the reviewer for the support of the publication of our manuscript on *Nature Communications*.